# HER2 inhibition increases non-muscle myosin IIA to promote tumorigenesis in HER2+ breast cancers

Samar M. Alanazi[1], Wasim Feroz[1], Rosalin Mishra[1], Mary Kate Kilroy[1], Hima Patel[1], Long Yuan[1], Sarah J. Storr[2], Joan T. Garrett[1]*

**1** James L. Winkle College of Pharmacy, University of Cincinnati, Cincinnati, OH, United States of America, **2** Nottingham Breast Cancer Research Centre, Division of Cancer and Stem Cells, School of Medicine, University of Nottingham, Nottingham, United Kingdom

* joan.garrett@uc.edu

**Data Availability Statement:** All relevant data are within the paper and its Supporting Information files.

## Abstract

HER2 is over-expressed in around 15% to 20% of breast cancers. HER3 plays a critical role in HER2 mediated tumorigenesis. Increased HER3 transcription and protein levels occur upon inhibition of HER2. We aimed to identify proteins that bound to HER3 upon inhibition of the HER family with the pan-HER inhibitor neratinib in HER2+ breast cancer cells. Immuno-precipitation of HER3 followed by mass spectrometry experiments found non-muscle myosin IIA (NMIIA) increased upon neratinib treatment relative to vehicle DMSO treatment. *MYH9* is the gene that encodes for the heavy chain of NMIIA. Breast cancer patients with high *MYH9* were significantly associated with a shorter disease specific survival compared to patients with low *MYH9* expression from the METABRIC cohort of patients. In addition, high MYH9 expression was associated with HER2+ tumors from this cohort. Immunoblots of whole cell lysates of BT474 and MDA-MB-453 HER2+ breast cancer cells demonstrated elevated HER3 and NMIIA protein levels upon neratinib treatment for 24 hours. To examine the role of NMIIA in HER2+ breast cancer, we modulated NMIIA levels in BT474 and MDA-MB-453 cells using doxycycline inducible shRNA targeting *MYH9*. *MYH9* knockdown reduces HER3 protein levels and concomitant reduction in downstream P-Akt. In addition, loss of MYH9 suppresses cell growth, proliferation, migration, and invasion. Our data reveals that NMIIA regulates HER3 and loss of NMIIA reduces HER2+ breast cancer growth.

## Introduction

HER2+ breast cancer is characterized by overexpression of human epidermal growth factor receptor 2 (HER2, HER2/neu, erbB-2). HER2 is an oncogene located on chromosome 17q12 and is amplified in about 20% of all breast cancers [1]. The HER family consists of four members: EGFR (ErbB1/HER1), ErbB2 (HER2), ErbB3 (HER3), and ErbB4 (HER4). HER2 always maintains an open confirmation with no known high affinity ligand and has potent kinase

**Funding:** The author(s) received no specific funding for this work.

**Competing interests:** The authors have declared that no competing interests exist.

activity, making it the ideal partner for dimerization [2]. HER3 plays a central role in HER2 + breast cancer cells. Studies have shown that blocking the kinase activity of HER2 results in upregulation of HER3 [3–5].

Approved HER2 targeted therapies include monoclonal antibodies that target the extracellular domain of HER2: trastuzumab, pertuzumab, and margetuximab (Margenza, margetuximab-cmkb), engineered for increased binding to activating Fcγ receptor IIIA (CD16A) and decreased binding to inhibitory Fcγ receptor IIB (CD32B) relative to trastuzumab [6, 7]. HER2 antibody-drug conjugates is another class of drugs targeting HER2 including trastuzumab emtansine (T-DM1), a trastuzumab-cytotoxic drug conjugate [8] and fam-trastuzumab deruxtecan-nxki (Enhertu), which delivers a topoisomerase I inhibitor [9]. Small molecules targeting the intracellular tyrosine kinase (TK) domain of HER2 that are FDA approved to treat HER2 + breast cancer are lapatinib, neratinib [10], and tucatinib [11]. These therapies have significantly improved the outcome of HER2+ breast cancer patients [10]. However, most HER2 + metastatic breast cancer patients will eventually develop resistance to HER2 targeted therapies [12]. Neratinib is one of the FDA approved HER2 inhibitors for extended adjuvant treatment of HER2+ breast cancer patients following adjuvant trastuzumab therapy [13].

Non-muscle myosin class II, (NMII) plays a variety of functions as a skeleton protein, such as involvement in cell motility, polarity, and adhesion. They also have a role in maintaining cell shape, and signal transduction. Non-muscle myosin IIA (NMIIA) is the myosin isoform consisting of non-muscle myosin heavy chain IIA (NMHCIIA), encoded by the *MYH9* gene, and regulatory and essential light chains that are shared with other NMII isoforms [14–17]. The *MYH9* gene is localized on chromosome 22q12.3. NMIIA belongs to a superfamily of motor proteins that promote mechanical forces when they bind to actin through hydrolysis of ATP. NMIIA is a hexameric molecule composed of a dimer of heavy chains, two regulatory light chains (RLC) and two essential light chains (ELC).

Several studies have shown that MYH9 has a critical role in cancer and demonstrated that NMIIA expression in cancer cells associated with progression of various types of cancers [18]. A study in MDA-MB-231 breast cancer cells showed that depletion of NMIIA with small-interfering RNA resulted in decreased MDA-MB-231 cell migration but enhanced lamellar spreading [19]. The overexpression of NMIIA was strongly correlated with esophageal squamous cancer and non-small cell lung cancer (NSCLC) progression and was associated with worse overall survival [20, 21]. In addition, NMIIA overexpression promotes invasion, lymph node metastasis in gastric cancer [22, 23], increases proliferation and migration in colorectal cancer (CRC) cells and pancreatic cancer [24, 25], and associated with poor prognosis in hepatocellular carcinoma (HCC) [26]. Collectively, the data of these studies indicate that overexpression of NMIIA could contribute to cancer progression and poor prognosis. In this study, we aim to investigate the role of NMIIA in HER2+ breast cancer cells. Our data indicate that NMIIA controls HER3 downstream signaling and promotes HER2+ breast cancer growth, proliferation, migration, and invasion.

## Materials and methods

### Cell culture and inhibitors

Human HER2+ breast cancer cells (BT474 and MDA-MB-453) were obtained from American Type Culture Collection, ATCC (Manassas, VA). Neratinib was obtained from LC Laboratories. BT474 cells were maintained in RPMI with 10% fetal bovine serum (FBS) at 37˚C in 5% $CO_2$ in a humidified incubator. MDA-MB-453 cells were maintained in DMEM with 10% fetal bovine serum at 37˚C in 5% CO2. Blebbestatin was obtained from SelleckChem (Cat # S7099). Doxycycline was obtained from MP Biomedicals, LLC (Cat # 198955)

## Immunoprecipitation

BT474 and MDA-MB-453 cells were grown to 70% confluency and lysed in 1% NP40 lysis buffer containing 20 mM Tris, pH 7.4, 150 mM NaCl, 10% glycerol, 1 mM EDTA, 1 mM EGTA, 5 mM NaPyro-PO4, 50 mM NaF, 10 mM β-glycerol-PO4. 0.7–1.4 milligrams (mg) of cleared lysate was incubated with 2 <g of HER3 (2F12) antibody (Invitrogen, Cat# MA5-12675), 2.3 <g IgG antibody (Invitrogen, Cat# MA1-10418), 1 <g NMIIA antibody (Abcam, Cat# ab238131) overnight at 4˚ C. The next day, 5 <l Protein A magnetic beads (Millipore) were added per 1 <g of antibody and incubated for 1 hr at 4˚C. Resuspended beads were washed with lysis buffer and boiled in SDS-PAGE loading buffer. The levels of HER3 and NMIIA from immunoprecipitated samples were analyzed by western blotting. For HER3 immunoprecipitates the recovered proteins were eluted and analyzed by silver staining, mass spectrometry, and western blotting.

## Silver staining, mass spectrometry

HER3 immunoprecipitated samples from BT474 treated with DMSO or neratinib for 24 hours were loaded in 4–20% SDS-polyacrylamide gel. The gel was stained using Pierce Silver Stain Kit (Thermo Scientific Cat# 24612). Briefly, gel was washed 2 × 5 minutes in ultrapure water, then fixed for 15 minutes in 30% ethanol:10% acetic acid solution. After fixation, gel was washed 2 × 5 minutes in 10% ethanol, then 2 × 5 minutes in ultrapure water. Gel sensitized for 1 minute, then washed 2 × 1 minute with water. Gel was stained for 30 minutes, then washed 2 × 20 seconds with ultrapure water. After staining and washing, the gel was developed for 2–3 minutes until appearance of bands. The selected bands were excised from the gel, pooled with like bands (resulting in 1 sample from the 24 hr neratinib and 1 sample from the DMSO), reduced with DTT, alkylated with IAA, and digested with trypsin. The peptides were extracted, dried by speed vac, resuspended in 10 ul of 0.1% FA and desalted using a micro C18 zip tip. The samples were mixed with alpha cyano 4-hydroxy cinnamic acid (HCCA) matrix and spotted for analysis by MALDI TOF-TOF. A MASCOT search of all entries was used to identify the proteins.

## Immunoblotting

BT474, MDA-MB-453 Cells were lysed in RIPA buffer (Thermo Fisher Scientific, Waltham, MA, USA, Cat#BP-115) supplemented with protease and phosphatase inhibitors (Thermo Fisher Scientific, Cat#88669). Immunoblotting was performed using specific primary antibodies (Cell Signaling Technology) for P-HER3 (Y1289) (Cat#4791), HER3 (Cat#12708), NMIIA (Cat#3403), P-Akt (T308) (Cat#13038), AKT (Cat#4691), P-Erk1/2 (Cat#4370), Erk1/2 (Cat#4695), and α- Tubulin (Cat#2125). Anti-rabbit IgG-horse radish peroxidase (HRP) was used as the secondary antibody (Santa cruz biotechnology sc-2357). α- Tubulin served as loading controls.

## *MHY9* expression in breast cancer patients

Gene expression microarray data from the Molecular Taxonomy of Breast Cancer International Consortium (METABRIC) breast cancer cohort were obtained from the CBioPortal for Cancer Genomics (n = 1903), with detailed description found in the original publication [27]. The gene expression data for MHY9 (mRNA expression z-values relative to diploid samples) were available from 1904 patients, with one breast sarcoma patient excluded from analysis (n = 1903). Statistical analysis was performed using IBM SPSS Statistics (version 28) and all differences were deemed statistically significant at the level of P≤0.05. X-tile was used to

determine cut-points for assessment using breast cancer specific survival [28]. Survival curves were plotted according to the Kaplan-Meier method with significance determined using the log-rank test. The Pearson $\chi 2$ test of association was used to determine the relationship between categorised protein expression and clinicopathological variables. Gene set enrichement analysis (GSEA) was performed using GSEA software [29, 30], with samples divided into MHY9 low and MHY9 high expression (https://www.gsea-msigdb.org/gsea/index.jsp) and gene enrichment determined in the curated hallmarks of cancer gene sets (h.all.v2022.1.Hs. symbols.gmt) [31, 32].

## qPCR analysis

Total RNA was extracted from BT474 and MDA-MB-453 cell lines using the RNA extraction mini kit (Qiagen, 74104, USA) according to the manufacturer's instruction. Two-step Real Time-qPCR was performed to assess the mRNA level of ERBB3 and MYH9. First strand cDNA was synthesized using iScriptTM cDNA Synthesis Kit (Biorad, 1708890, USA). qPCR was set up using CFX96 Real-Time System (Biorad, USA). GAPDH was used as an internal control. Relative ERBB3 and MYH9 mRNA expression was presented by $2{-}\Delta\Delta CT$ method. Paired primer sequences used for ERBB3: 5`-GGGTTAGAGGAAGAGGATGTCAAC-3` (forward) and 5`-GGGAGGAGGGAGTACCTTTGAG-3` (reverse); for MYH9: 5`-GTGAGAAG GAGACAAAGGCG-3` (forward) and 5`-GACTTCTCCAGCTCGTGGAC-3` (reverse); and for GAPDH: 5`-AAGAAGGTGGTGAAGCAG-3` (forward); 5`-TCATACCAGGAAATG AGC-3`.

## Generation of inducible *MYH9*-deficient cells

To ablate MYH9 expression, we transduced BT474 and MDA-MB-453 with doxycycline-dependent inducible lentiviruses expressing shRNAs that target MYH9 expression (GE Dharmacon, Lafayette, CO). A non-targeting shRNA lentivirus was used to generate control cell lines. The inducible lentiviral shRNA vector, which uses the Tet-On inducible system, only allows the expression of either MYH9-targeting shRNA (Construct (1) AGTGAGGACGA GCTACTCT, Construct (2) ATTGATACCCAAGAGATGG, Construct (3) TGTTGGGAGAAG CCACTGG or a nontargeting shRNA when cells are treated with doxycycline. The expression of GFP is driven by the same tetO promoter so that the transduction and shRNA expression upon doxycycline treatment can be visually tracked by green fluorescence. Briefly, BT474 and MDA-MB-453 cells were seeded in 96-well tissue culture plates. The following day, serum free media containing viral particles was added and incubated for 16 h; then, the medium containing viral particles was replaced with fresh media supplemented with 10% FBS. To select transduced cells, medium containing 1–2 μg/ml puromycin was added, and the cells were maintained for three days. Because the inducible lentiviral vector contains a puromycin-resistance gene, transduced cells survive the selection by puromycin, while cells without the integration of lentiviral DNA are killed by puromycin. To evaluate whether puromycin selection was complete, 500 ng/ml doxycycline (MP Biomedicals, LLC) was added for 48 h to induce the expression of shRNAs and GFP. The efficiency of inhibiting MYH9 expression by shRNA was determined by immunoblotting.

## Matrigel colony formation

Three-dimensional (3D) growth assays were performed in growth factor-reduced matrigel (BD Biosciences, San Jose, CA, USA) where 96 well plates were coated with 80 <L of matrigel/ well. BT474, MDA-MB-453 shCTRL or shMYH9 cells ($1 \times 10^4$/well) were plated and incubated at 37°C for 24 h. Cells were treated with vehicle (DMSO), 500 nM doxycycline or 200 nM neratinib or combination of both every alternate day for doxycycline. After 10 days of

incubation, colonies were visualized, and photographs were captured from 3 random fields under microscope at 10× magnification. The areas of the colonies were measured by ImageJ and represented as mean areas normalized to DMSO control.

## MTT cell proliferation assay

Both BT474 and MDA-MB- 453 shCTRL or shMYH9 cells ($2 \times 10^4$/well) were plated in 96 well plates in triplicate and incubated at 37˚C for 24 h. Cells were treated with vehicle (DMSO), 500 nM doxycycline or 200 nM neratinib or combination of both. After 72 hr, the media containing the drug was replaced with 5 mg/mL MTT (3-(4,5-dimethylthiazol- 2-yl)-2,5-diphenyltetrazolium bromide) dissolved in cell line specific media and incubated for 4 hr. After 4 hr the media was aspirated, and crystals were dissolved with isopropanol (Molecular Grade, Fisher BioReagents). The absorbance was read at 570 nm using a microplate reader (SPECTR Amax PLUS Microplate Spectrophotometer Plate Reader, Molecular Devices Corporation). Data is represented as the mean of at least two independent experiments ± SEM.

## Spheroid cultures and PrestoBlue assay

Spheroids were established by seeding cells in a 96-well Ultra-Low Attachment plate (Corning) in 200 <L volume per well. Extracellular matrix (ECM)-rich Cultrex (Trevigen Inc.) was included at a final concentration of 100 <g/mL to aid in the formation of spheroids. Cells were treated upon plating and every 48 hours with 500 nM doxycycline or vehicle. After centrifugation for at least 5 min at 300 Relative Centrifugal Field (RCF), the cultures were maintained in a cell culture incubator at 37˚C with 5% $CO_2$ in a humidified atmosphere. Spheroids were incubated with 10% (v/v) PrestoBlue (Thermo Fisher Scientific) for 30 min in the cell culture incubator, and the relative fluorescence (RFU) at different time points and measured utilizing a plate reader (Fluostar Omega, BMG Labtech; FlexStation II, Molecular Devices) at excitation and emission wavelengths of 544 nm and 590 nm, respectively. Data shown is normalized to the values for vehicle treatment.

## Cell migration and invasion assay

BT474 and MDA-MB- 453 shCTRL or shMYH9 cells were seeded at a density of $5 \times 10^4$ cells/well in 6 well plates and treated vehicle (DMSO), 500 nM doxycycline or 200 nM neratinib or combination of both. After 24 h, cells were counted and $2 \times 10^4$ cells/well were added to the upper chamber (for invasion the champers were coated with matrigel) of transwell and incubated for 48 h. After 48 h, migrated and invaded cells in the lower chamber were stained with 0.5% crystal violet and images were captured from three areas under phase contrast microscope. The intensities from three areas collected from three independent experiments were measured using ImageJ and expressed as % of control and represented graphically.

## Small-interfering RNA

4x106 BT474 and MDA-MB-453 cells were transfected with siRNAs specifically targeting HER3 (siHER3) or control siRNA (siCon). To knock down HER3 expression, siRNA against a HER3 target sequence ACCACGGTATCTGGTCATAAA was used (Dharmacon International). Mismatched siRNA with a target sequence of GGAAGC AGACTCACTCTTATA was used as a negative control.

## Statistical analysis

Data are shown as the mean ± standard error of mean (SEM) and representative of at least three independent experiments unless indicated otherwise. Statistical analysis was performed by two sample paired 't'-test using ANOVA (Graph Pad Prism 7). The data was considered statistically significant if p<0.05.

## Results

### Enhanced NMIIA–HER3 levels in response to HER2 inhibition in HER2 + breast cancer cells

Previous studies have shown that inhibition of HER2 results in upregulation of HER3 [3]. We sought to identify HER3 binding partners in the presence or absence of pharmacological inhibition of the HER family using the irreversible pan HER inhibitor neratinib. Mass spectrometry experiments were performed to identify HER3 binding proteins in HER2+ BT474 breast cancer cells. NMIIA was increased upon inhibition of HER2 with neratinib relative to DMSO control treatment from HER3 immunoprecipitates (**Fig 1A**). Immunoprecipitation experiments were performed in BT474 and MDA-MB-453 breast cancer cells using a HER3, IgG control, or NMIIA antibody. Immunoblots performed using antibodies against NMIIA and HER3 showed increased NMIIA and HER3 levels upon treatment with 200nM neratinib for 24 hours in HER3 and NMIIA immunoprecipitates. 200 nM of neratinib was chosen for this and subsequent experiments as this is a clinically achievable concentration in patients [33]. We observed 24 hours of neratinib treatment resulted in enhanced NMIIA bound to HER3 compared to vehicle treated cells (**Fig 1B and 1C**).

### Significance of NMIIA in breast cancer

We examined the gene expression of MHY9 in the METABRIC cohort and observed that patients with high MHY9 expression have shorter survival to patients with low MHY9 levels (**Fig 2A**). We next examined if MHY9 expression was associated with survival in patient subgroups; high MHY9 expression was significantly associated with adverse survival of patients with mutations in P13KCA mutations ($P = 0.007$) but was not associated with survival of patients with PI3KCA wild type tumors ($P = 0.558$) (**Fig 2B and 2C**). PI3KCA is the most

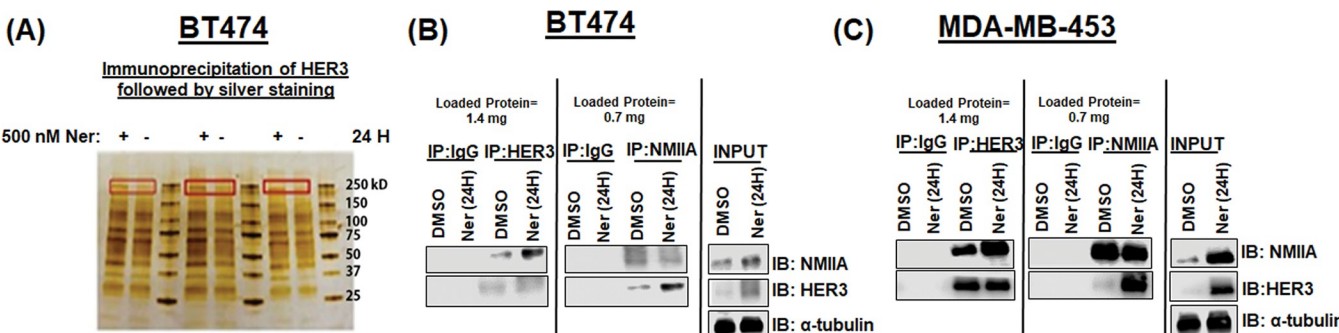

**Fig 1. Enhanced NMIIA–HER3 levels upon HER2 inhibition in HER2+ breast cancer.** (**A**) BT474 cells were treated with DMSO or 500 nM neratinib for 24 hours. Cells were lysed and an immunoprecipitation for HER3 was performed. Triplicate samples are shown. Bands in red represent NMIIA as determined by mass spectrometry. (**B**) BT474 cells were treated with 200 nM neratinib or vehicle DMSO for 24 hours. The cells were lysed using NP-40 buffer and immunoprecipitated using a HER3, IgG, or NMIIA antibody. The products were analyzed by 10% SDS-PAGE followed by NMIIA and HER3 immunoblots. (**C**) MDA-MB-453 cells were treated with 200 nM neratinib and DMSO for 24 hours. The cells were lysed using NP-40 buffer and immunoprecipitated using a HER3, IgG, or NMIIA antibody. The products were analyzed by 10% SDSPAGE followed by NMIIA and HER3 immunoblots.

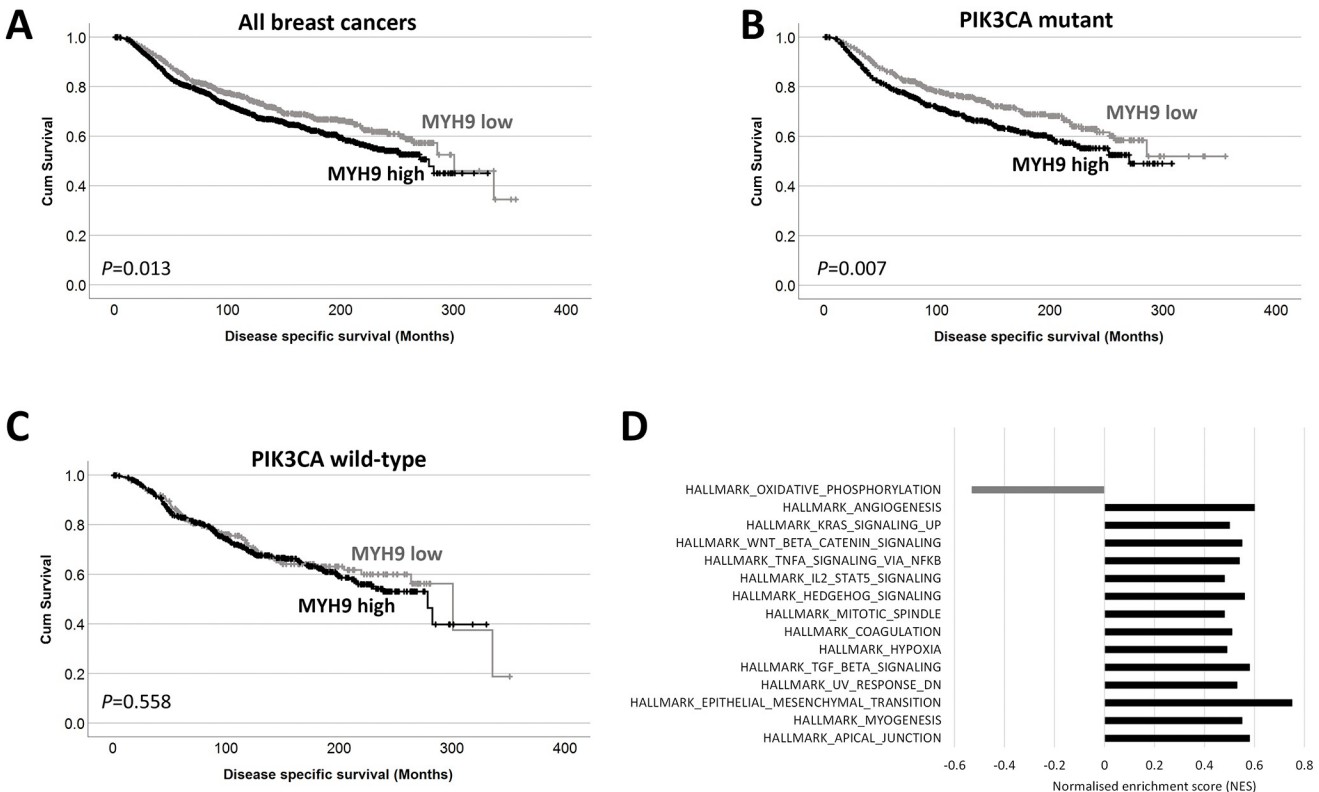

**Fig 2. MYH9 is associated with breast cancer survival.** Kaplan-Meier analysis of MHY9 expression where the impact of low (grey line) and high (black line) expression are shown (A). The impact of low (grey line) and high (black line) MHY9 expression are shown in PI3KCA mutant tumors (B) and in wild type tumors (C). GSEA results showing normalized enrichment scores for the gene sets enriched as a FDR of 1% in MHY9 low samples (gray) and MHY9 high samples (black).

frequently mutated gene in the METABRIC cohort, and of those patients with matched gene expression data, 41.8% of cases are PI3KCA mutant (795/1903).

High MHY9 expression was associated with progesterone receptor negative tumors ($\chi^2$ = 19.970, d.f. = 1, $P<0.001$), HER2 positive tumors ($\chi^2$ = 12.440, d.f. = 1, $P<0.001$), ER negative tumors ($\chi^2$ = 41.608, d.f. = 1, $P<0.001$), patients under 40 years ($\chi^2$ = 4.885, d.f. = 1, $P$ = 0.027), and tumors with p53 mutations ($\chi^2$ = 23.007, d.f. = 1, $P<0.001$); data is shown in **Table 1**. GSEA was used to explore METABRIC microarray data for enrichment of genes in the curated hallmarks of cancer gene sets in high MHY9 tumors. Normalized enrichment scores (NES) with a false discovery rate of 1% identified 14/50 enriched gene sets in MHY9 tumors, compared with 1/50 in MHY9 low tumors (**Fig 2D**).

## HER3 and NMIIA protein levels are increased upon inhibition of HER2 in HER2+ breast cancer

We next sought to examine overall levels of HER3 and MYH9 mRNA and protein upon treatment with neratinib in HER2+ breast cancer cell lines. Levels of HER3 and MYH9 mRNA in BT474 and MDA-MB-453 were assessed using qRT-PCR. We did not observe that mRNA levels of HER3 or MYH9 consistently changed upon HER2 inhibition with neratinib (data not shown). We found that 24 hours of neratinib treatment reduced P-HER2 and concomitant reduction in total HER2 protein as has been reported [34, 35]. We found that neratinib treatment increased HER3 and NMIIA protein levels (**Fig 3**).

**Table 1. Associations between MHY9 expression and clinicopathological variables.**

| Variable | | MHY9 expression | | |
|---|---|---|---|---|
| | | Low | High | *P* value |
| P53 mutation status | Wild type | 487 (25.7%) | 754 (39.7%) | <0.001 |
| | Mutant | 185 (9.7%) | 472 (24.9%) | |
| PgR status | Negative | 269 (14.2%) | 622 (32.8%) | <0.001 |
| | Positive | 403 (21.2%) | 604 (31.8%) | |
| HER2 status | Negative | 613 (32.3%) | 1050 (55.3%) | <0.001 |
| | Positive | 59 (3.1%) | 176 (9.3%) | |
| ER status | Negative | 100 (5.3%) | 343 (18.1%) | <0.001 |
| | Positive | 572 (30.1%) | 883 (46.5%) | |
| Patient age | 40 years or less | 30 (1.6%) | 86 (4.5%) | 0.027 |
| | Greater than 40 years | 639 (33.8%) | 1136 (60.1) | |
| Tumor size | 2cm or less | 297 (15.8%) | 521 (27.7%) | 0.616 |
| | Greater than 2cm | 373 (19.9%) | 687 (36.6) | |

The **P** values are resultant from Pearson χ2 test of association.

## Inducible MYH9 deficient HER2+ breast cancer cells demonstrate that NMIIA controls HER3 signaling pathway

Three different MYH9-targeting shRNAs were designed and evaluated for their abilities to inhibit MYH9 expression. shMYH9 (1) was found to have the highest inhibitory efficiency in both cell lines and was used to generate MYH9 deficient BT474 and MDA-MB-453 cells (**Fig 4A**). We next examined the effect NMIIA loss has on HER3 signaling in the absence or presence of neratinib. Lentivirally transduced MDA-MB-453 and BT474 cells with shMYH9 and doxycycline induction demonstrate a reduction in HER3 protein levels compared to cells lentivirally transduced with a control sequence and doxycycline induction (**Fig 4B**). Strikingly, we observed a concomitant reduction in P-HER3 (Y1289) and downstream P-Akt (T308) and P-Erk1/2. Y1289 of HER3 resides within a YXXM motif and participates in signaling to PI3K upon phosphorylation, activating downstream signaling [36]. Neratinib alone reduced P-HER3, P-Akt and P-Erk1/2. The combination of doxycycline to induce MYH9 knockdown and neratinib treatment resulted in further reduction in P-HER3, P-Akt and P-Erk1/2. Furthermore, we investigated the effect of NMIIA siRNA in the presence or absence of neratinib in BT474 and MDA-MB-453 cells (**S1 Fig**). Notably, we observed NMIIA RNA and proteins levels are increased upon neratinib treatment in cells transfected with control siRNA. Cells transfected with MYH9 siRNA in the absence of neratinib treatment had decreased MYH9 mRNA and protein levels. MYH9 siRNA reduced HER3 mRNA and proteins levels in the absence of neratinib treatment, indicating that mechanism(s) exist for NMIIA to modulate HER3 mRNA and protein levels. Furthermore, in cells transfected with MYH9 siRNA, neratinib treatment increased both HER3 and NMIIA RNA and protein levels compared to vehicle DMSO treatment. Additionally, we used the pharmacological inhibitor for NMII blebbistatin to evaluate the mRNA and protein levels of NMIIA and HER3. Blebbistatin inhibits myosin ATPase activity [37]. It binds halfway between the nucleotide binding pocket and the actin binding cleft of myosin, predominantly in an actin detached conformation [38]. It has been shown that blebbistatin reduces NMIIA protein levels [39]. Blebbistatin treatment results in reduced HER3 mRNA and protein levels (**S2 Fig**).

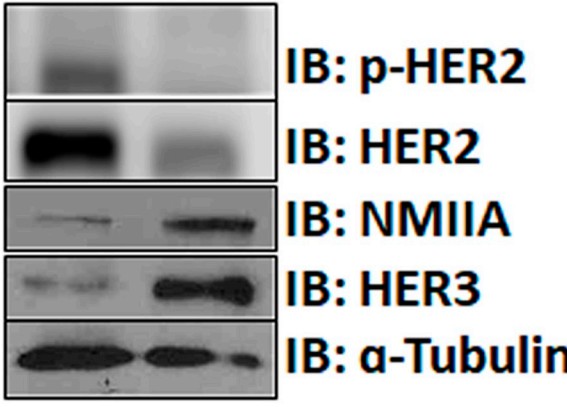

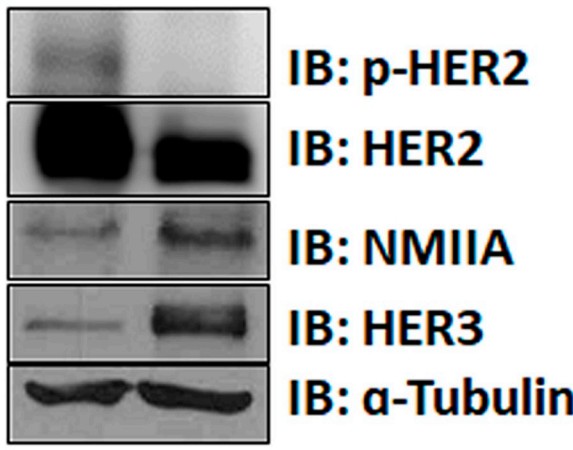

**Fig 3. HER3 and NMIIA protein levels are increased upon inhibition of HER2 in HER2+ breast cancer cells.**
BT474 and MDA-MB-453 cells were treated with vehicle or neratinib (200 nM) for 24 hours. Cells were lysed and
products analyzed by 10% SDS-PAGE followed by probing immunoblots with antibodies for p-HER2, HER2, NMIIA,
HER3 and α-tubulin.

## MYH9 knockdown and neratinib treatment inhibit matrigel colony formation, and proliferation in HER2+ breast cancer cells

Examination of BT474 and MDA-MB-453 cells transduced with a doxycycline inducible construct expressing shRNA targeting MYH9 grown on a basement membrane of matrigel indicated that combination doxycycline (500 nM) and neratinib (200 nM) inhibit colony formation better than either single knockdown of MYH9 or neratinib treatment. Our data showed that the combination of doxycycline with neratinib significantly suppressed colony

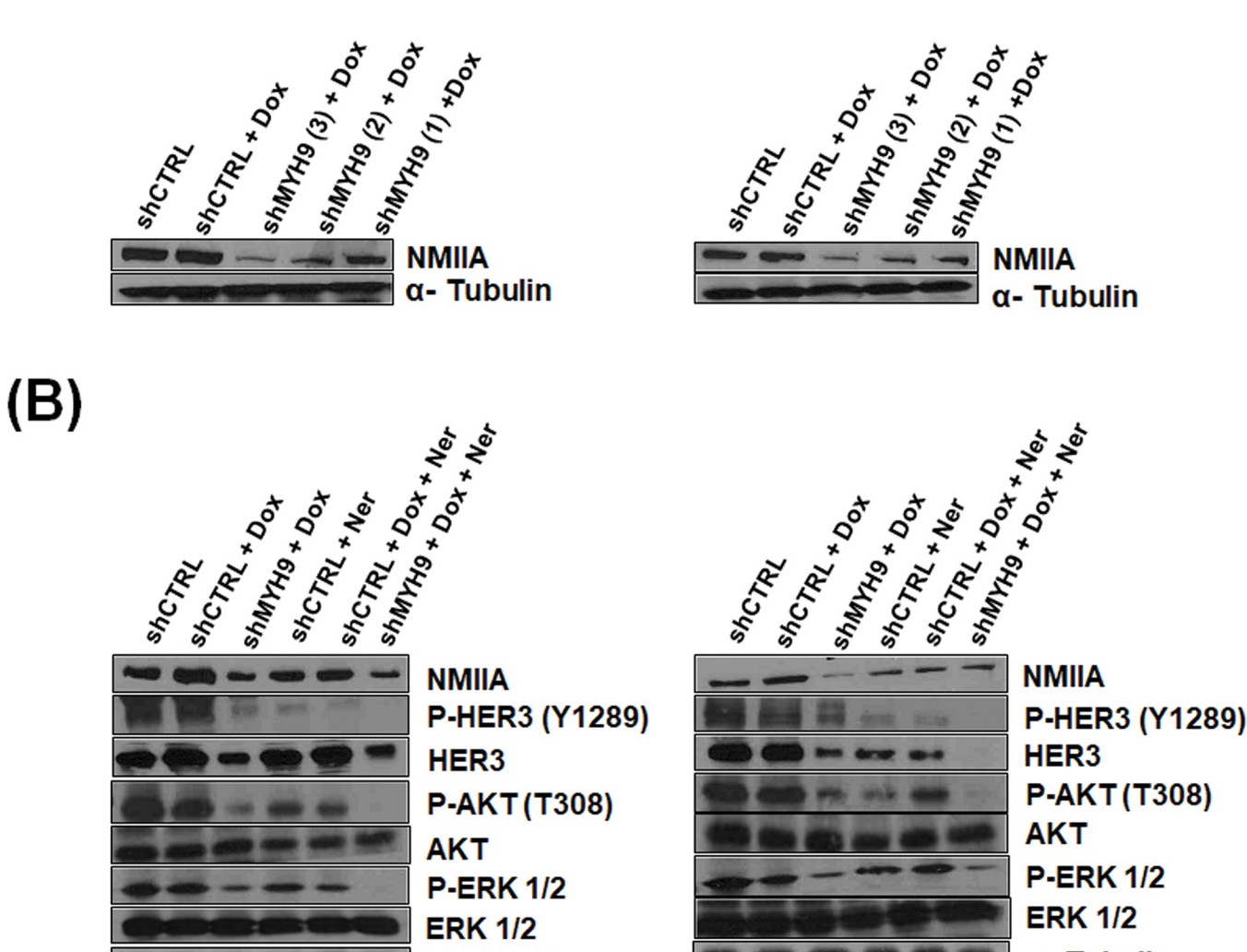

**Fig 4. Generation of HER2+ breast cancer cells lentivirally transduced with shRNA targeting MYH9: Loss of NMIIA inhibits HER3 signaling axis. (A)** Western blot showing MYH9 expression in transduced and selected BT474 and MDA-MB-453 cells after 48-hour of 500 nM doxycycline induction. **(B)** Western blot using the indicated antibodies in transduced and selected BT474 and MDA-MB-453 cells after 48-hour of 500 nM doxycycline induction and/or 24-hour of 200 nM neratinib treatment as indicated.

formation on matrigel compared to individual treatment or DMSO control in BT474 and MDA-MB-453 cells transduced with shMYH9. BT474 cells were more sensitive to the combination of doxycycline and neratinib marked by significantly reduced colonies versus single treatment of doxycycline, neratinib, or vehicle DMSO (**Fig 5A and 5B**). We speculated that the combination of doxycycline-induced MYH9 knockdown and neratinib could effectively inhibit proliferation of BT474 and MDA-MB-453 cells transduced shMYH9. Accordingly, BT474 and MDA-MB-453 cells transduced with shRNA targeting MYH9 were treated with doxycycline (500 nM) and neratinib (200 nM) for 72 hr. After 72 hr, the media containing the drug was replaced with 5 mg/mL MTT (3-(4,5-dimethylthiazol-2-yl)-2,5-diphenyltetrazolium bromide) dissolved in cell line specific media and incubated for 4 hr. After 4 hr the media was

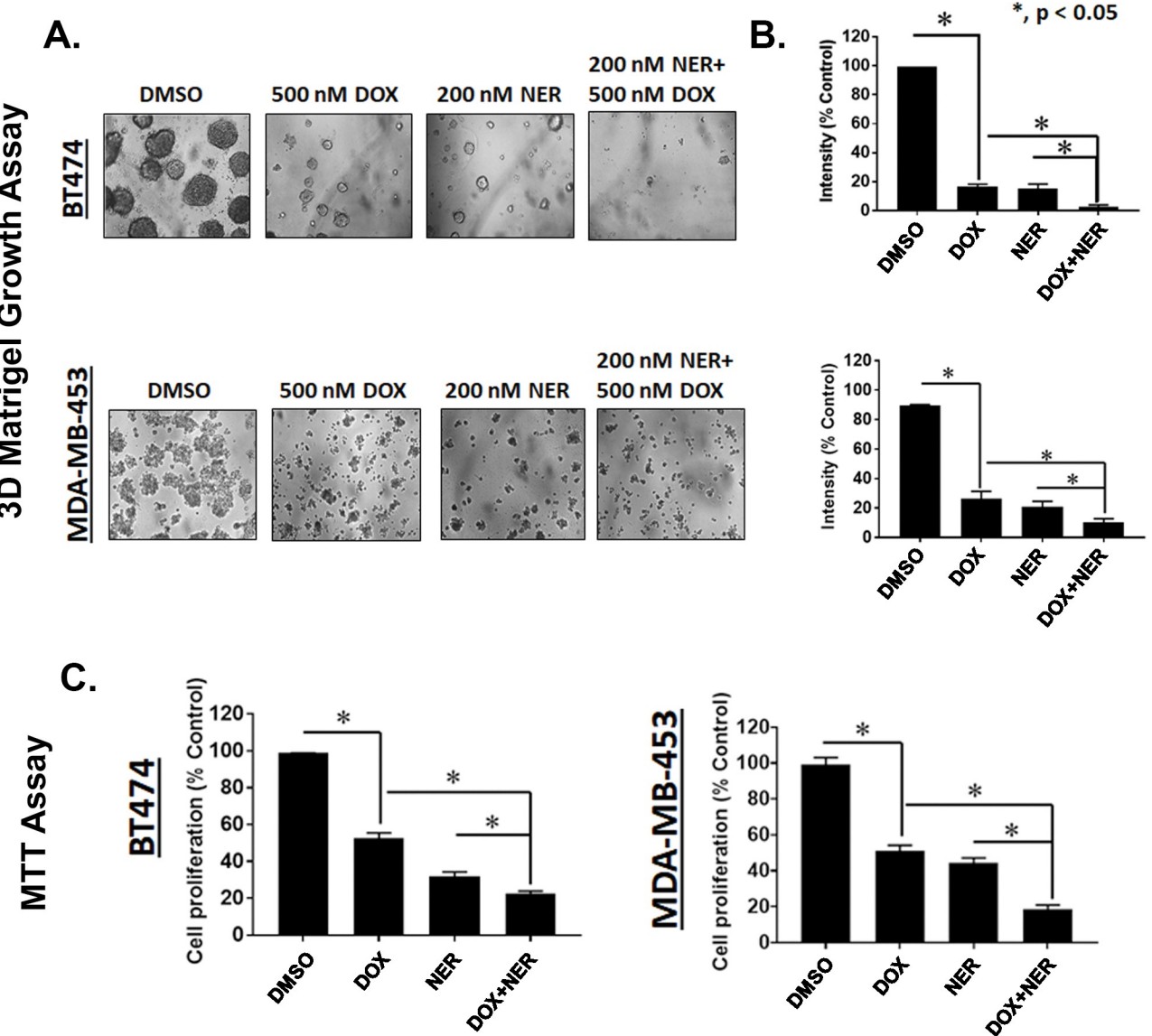

**Fig 5. MYH9 knockdown and neratinib treatment inhibit colony formation on Matrigel and proliferation in HER2+ breast cancer cells. (A)**
BT474 and MDA-MB-453 cells transduced with MYH9 shRNA were seeded on a basement membrane of matrigel at a density of $2 \times 10^4$ cells/well.
Cells treated with vehicle (DMSO), doxycycline (500 nM)/48 h, neratinib (200 nM) once, and combination. Phase contrast images of acini for all cell
lines were captured at 10× magnification and the average size of each cellular structure was quantified and expressed as mean of areas ± SEM, n = 5
random fields * p <0.05 for indicated comparison (**B**). (**C**) BT474 and MDA-MB-453 cells transduced with a doxycycline inducible shRNA targeting
MYH9 were treated with doxycycline (500 nM) and neratinib (200 nM) for 72 hr. After 72 hr, the media containing the drug was replaced with 5 mg/
mL MTT dissolved in cell line specific media and incubated for 4 hr. After 4 hr the media was aspirated, and crystals were dissolved with isopropanol.
The bar graphs are represented as mean; Error bars: SEM, n = 3 * p <0.05 for indicated comparison.

aspirated, and crystals were dissolved with isopropanol. We observed that the combination of
neratinib and doxycycline to deplete MYH9 in BT474 and MDA-MB-453 cells resulted in a
statistically significant reduction in proliferation compared to single treatment doxycycline or
neratinib in both cell lines (Fig 5C). Parallel experiments in BT474 and MDA-MB-453 cells
expressing control non-targeting shRNA were performed. For cells grown on a basement
membrane of Matrigel, doxycycline added to neratinib did not result in a further reduction of
colony size (S3A and S3B Fig). In BT474 cells transduced with control shRNA we observed a

further reduction in proliferation for the combination of doxycycline and neratinib compared to neratinib treatment alone which is unexpected as we have found that cells stably expressing lentivirus tolerate 500 nM doxycycline (S3C Fig). We investigated the growth of BT474 cells expressing shCTRL or shMYH9 in spheroids by plating 1000 cells in ultra-low attachment plates. Spheroids can recapitulate the basic 3D structure of tumors and more closely mimic cancer cells *in vivo*. On day 1, 3 and 5 cells were treated with vehicle or doxycycline. Images of spheroids taken on day 3, 5 and 7 revealed BT474 cells expressing shMYH9 treated with doxycycline were smaller compared to vehicle treatment. Furthermore, assessing cell viability using PrestoBlue reagent revealed statistical reductions in cell viability on days 3, 5 and 7 in doxycycline treated cells versus vehicle treated cells (S4 Fig). Doxycycline treatment had no effect in BT474 cells expressing shCTRL.

### MYH9 knockdown and neratinib treatment suppress HER2+ breast cancer cell migration and invasion

Knockdown of MYH9 in BT474 and MDA-MB-453 cells reduced migration and invasion of cells. We assessed migration by plating cells in the upper chamber of transwell plates using 10% FBS as a chemoattractant in the lower chamber. BT474 and MDA-MB-453 cells transduced with shMYH9 and treated with doxycycline (500 nM) and neratinib (200 nM) had a significantly lower number of migrated cells compared with single agent doxycycline or neratinib in both cell lines (Fig 6A and 6B). We next assessed invasion via coating the upper chamber with a layer of matrigel requiring cells to travel through the matrigel to enter the lower chamber. We observed that BT474 and MDA-MB-453 cells containing shRNA targeting MYH9 treated with doxycycline (500 nM) and neratinib (200 nM) had a significantly lower number of invading cells compared with single agent doxycycline or neratinib in both cell lines (Fig 6C and 6D). Neratinib treatment reduced migration and invasion of BT474 and MDA-MB-453 cells expressing control non-targeting shRNA (S5 Fig). Doxycycline alone or added to neratinib did not affect migration or invasion of cells expressing non-targeting shRNA.

## Discussion

Previous studies indicate that inhibition of HER2 TK activity results in HER3 upregulation [40]. In our study we sought to identify HER3 binding proteins upon inhibition of HER2 by the irreversible HER inhibitor neratinib. We identified NMIIA encoded by MYH9 as a HER3 binding protein upon pharmacological inhibition of HER2 in HER2+ breast cancer cells (Fig 1).

NMIIA is known to be involved in many cellular functions including cell adhesion, cytokinesis, polarization, division, and migration [41–43]. It is also a key regulator in human diseases such as cancer [18, 44]. NMIIA can have an oncogenic role in different types of cancer such as triple negative breast cancer, esophageal squamous cancer, NSCLC, gastric cancer, CRC, pancreatic cancer, HCC and nasopharyngeal carcinoma [19–26, 45]. Recent work to delineate how NMII isoforms function upon a stiffening extracellular matrix in cancer cells found that NMIIA aides the establishment of cell polarity through actin polarization at the cell leading edge, lamellipodia formation and focal adhesion turnover [46].

Conversely, other studies have shown that NMIIA can act as a tumor suppressor. Repression of MYH9 predisposed mice to squamous cell carcinoma (SCC), as MYH9 was identified from an RNA interference screen [47]. The mechanism by which NMIIA acts as a tumor suppressor was via a noncanonical role in nuclear retention of activated p53. When NMIIA is defective, even though the p53 pathway can be induced in response to DNA damage, it did not occur because of an inability of p53 to remain and/or accumulate in the nucleus. Additional

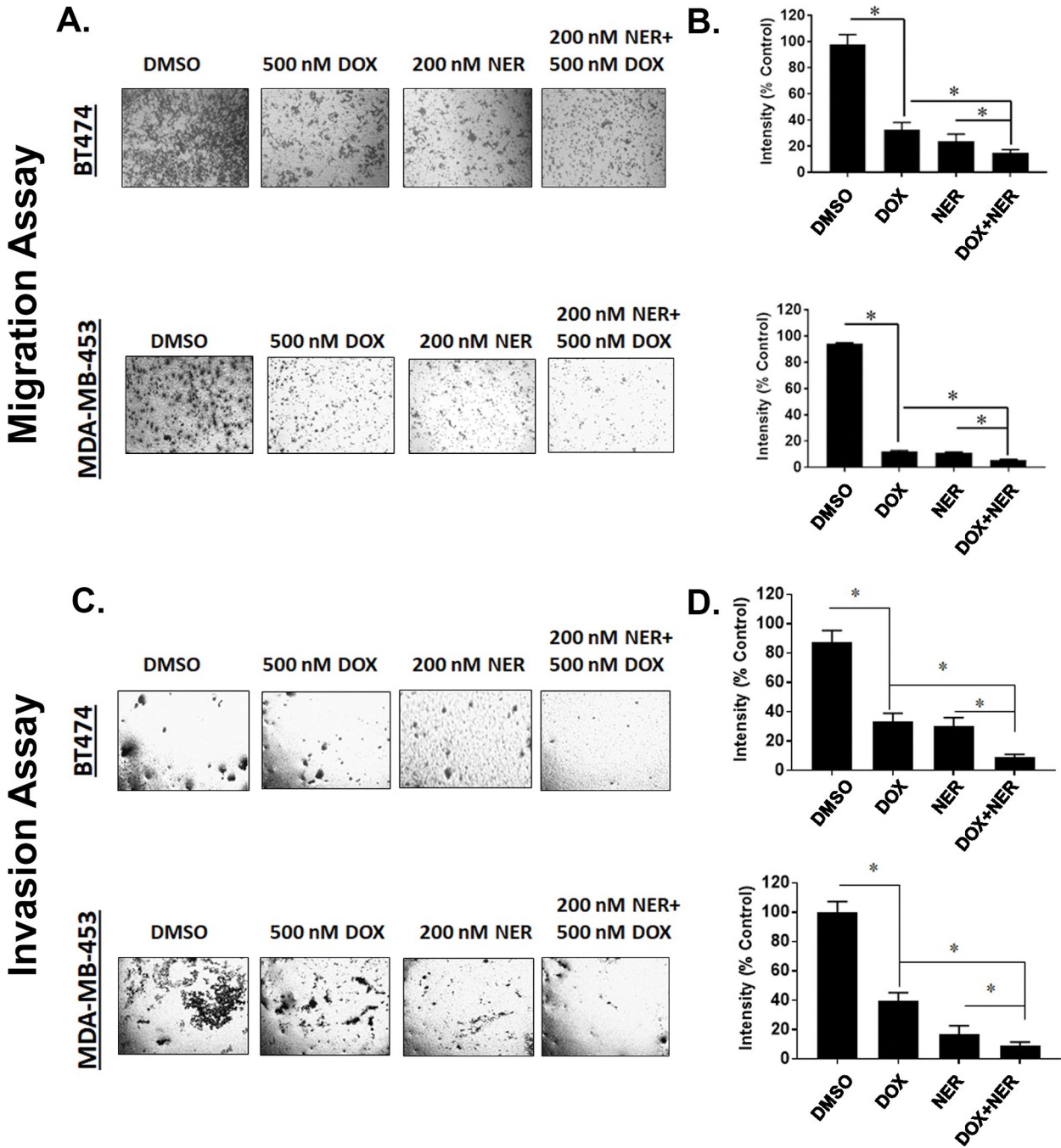

**Fig 6. MYH9 knockdown and neratinib treatment inhibit migration and invasion in HER2+ breast cancer cells. (A)** BT474 and MDA-MB-453
($5 \times 10^4$ cells/well) stable cells transduced with MYH9 shRNA were seeded in six well plates and treated with vehicle (DMSO), doxycycline (500
nM), neratinib (200 nM) and indicated combination in serum free media for 24 h. Post 24 hr treatment, these treated cells ($2 \times 10^4$ cells/well) were
added to the upper chamber of transwell plates with 10% FBS as chemoattractant in the lower chamber. After 24h, the migrated cells were stained
and captured. **(B)** The intensities of the migrated cells were measured using ImageJ and expressed as mean of % control and represented as bar
graph. Error bars: SEM, n = 4, * p <0.05 for indicated comparison. **(C)** Invasion assays were performed where each chamber was coated with 20 <l
matrigel mixed with 80 <l serum free media and incubated at 37˚ for 30 mins. Stable cells (BT474 and MDA-MB-453) expressing MYH9 shRNA
were seeded at a density of $5 \times 10^4$ cells/well and treated and incubated as above. After 24h, the invaded cells were stained and photographed **(D)**
The intensities of the invaded cells were measured using ImageJ and expressed as mean % of control. Error bars: SEM, n = 3* p <0.05 for indicated
comparison.

work found that low MYH9 expression is associated with decreased survival in patients with head and neck cancer harboring stratified low-risk mutant p53 but not high-risk mutant p53 [48]. More than half of all human cancers contain a TP53 mutation. Notably BT474 cells have been reported to have a p53 mutation and MDA-MB-453 cells have a deletion in p53 [49]. As the breast cancer models in our study may not depend on functional p53, the role of NMIIA in nuclear retention of activated p53 may be irrelevant for the breast cancer models in our study. Mice with homozygous deletion of NMIIA consistently developed SCC of the tongue [50]. Interestingly genetic deletion of NMIIA did not affect the stability of p53 but was ascribed to NMIIA maintaining mitotic stability during cell division. An additional study found that genetically knocking down MYH9 in mouse melanoma cells increased growth and metastasis *in vivo* in mouse models which was attributed to increased epithelial to mesenchymal (EMT) markers and potentially increased angiogenesis and inflammatory cell recruitment [51]. These studies point to the context in which NMIIA may be needed throughout development and the interplay with the tumor microenvironment in determining if NMIIA permits or hampers tumor growth.

Blebbistatin reversibly inhibits NMII by blocking its ATPase activity [37, 38, 52, 53]. Due to blebbistatin's non-specificity and cytotoxicity [54], it may not be a good candidate to target NMIIA in HER2+ breast cancer. Recently however blebbistatin has been used *in vivo* in tumorgenicity studies demonstrating reduced tumor volume in a dose-dependent manner in a NSCLC model [39]. Toxicity or weight loss with blebbistatin in mice was not reported although the lack of specificity for blebbistatin is a major consideration. Blebbistatin has been reported to inhibit NMIIA and NMIIB along with several striated muscle myosins and data indicate that blebbistatin inhibits arterial smooth muscle tonic contractions [55, 56]. There is a strong rationale for the development of more specific inhibitors of NMIIA that could be used in cancer treatment, potentially in HER2+ breast cancers from data presented in this study.

Recently, Orgaz et al. demonstrated that myosin II activity increases during melanoma adaptation to MAPK pathway inhibition. High myosin II activity correlated with targeted and immunotherapy resistant melanomas [57]. Rho-kinase (ROCK) is upstream of NMIIA. ROCK inactivates the myosin light chain 2 (MLC2) phosphatase, which leads to increased phosphorylation of MLC2 (p-MLC2) and myosin II activity. ROCK inhibitors improve the efficacy of MAPK inhibitors and immunotherapies in melanoma models [57]. The utility of ROCK inhibitors should be considered in combination with HER2 inhibition in HER2+ breast cancers given the paucity of NMIIA specific inhibitors.

We observed a bidirectional relationship between NMIIA and the HER family. **Figs 1** and **3** demonstrate that inhibition of HER signaling with neratinib treatment increases NMIIA protein levels. Future studies will define the mechanistic details of neratinib induced MYH9 upregulation. There are limitations to our work. This study does not determine mechanistically how HER3 and NMIIA interact nor the mechanism(s) by which loss of NMIIA reduces HER3 protein levels or how the loss of HER3 reduces NMIIA protein levels. The immunoprecipitation experiments shown in Fig 1B and 1C are challenging to interpret. The immunoprecipitation of HER3 followed by immunoblotting of HER3 reveals unequal amounts of HER3 between lanes, similarly for immunoprecipitation of NMIIA followed by immunoblotting of NMIIA. As shown in Fig 3, 24 hours of neratinib treatment results in increased NMIIA and HER3, correlating with the increased amount of HER3 and NMIIA detected from the immunoprecipitation and immunoblotting experiments shown in Fig 1B and 1C. Future studies will mechanistically decipher which compartment(s) of the cell NMIIA and HER3 interact and if blebbistatin can inhibit the interaction. Future studies will delineate if loss of MYH9 affects the half-life of HER3 protein and examine the effects of overexpression of NMIIA.

We have found NMIIA is increased upon HER2 kinase inhibition. HER2 inhibition in the clinic is very rarely curative to women with HER2+ metastatic breast cancer. As such this work aims to determine if there is potential validity in co-targeting NMIIA and HER2. Cumulatively, we found that BT474 and MDA-MB-453 cells containing shRNA targeting MYH9 treated with doxycycline and neratinib had a statistically significant: (1) reduction in the size of colonies grown on a basement membrane of Matrigel, (2) reduction in proliferation, (3) reduction in migrating cells and (4) lower number of invading cells compared with single agent doxycycline or neratinib in both cell lines. These experiments support further examination of potentially co-targeting NMIIA and HER2 in HER2+ breast cancer. Future studies in HER2+ breast cancers would determine conditional loss of MYH9 in HER2+ breast cancer xenografts *in vivo* and additionally examine the efficacy of inhibitors of ROCK, upstream of NMIIA, in combination with HER2 inhibition.

## Conclusions

This study shows a bidirectional relationship between NMIIA and the HER family in HER2 + breast cancer cells. HER kinase inhibition increases NMIIA. Loss of NMIIA reduces HER3 protein and concomitant reductions in PI3K/Akt and MAPK signaling. Loss of NMIIA in combination with HER kinase inhibition results in reduction in HER2+ breast cancer cell proliferation, growth on a basement membrane, migration and invasion. This study identifies NMIIA as a potential therapeutic target in HER2+ breast cancers.

## Supporting information

**S1 Fig. NMIIA knockdown reduces HER3 and NMIIA expression that is rescued by inhibition of HER2 with 24h treatment of neratinib in HER2+ breast cancer cell lines.** BT474 and MDA-MB-453 were seeded in 60 mm plates ($2x \ 10^6$ cells/plate), and forward and reverse transfection were performed using siRNA targeting MYH9 or scrambled sequence (control) and lipofectamine RNAmax mixture for 48 hours. **(A)** BT474 and MDA-MB-453 cells were then treated with 200 nM neratinib or DMSO for 24 hours. RNA was isolated and MYH9 and HER3 RNA levels determined using real time PCR. **(B)** BT474 and MDAMB-453 cells were treated with neratinib (200 nM) for 0–24 hours. Cells were lysed and products analyzed by 10% SDS-PAGE followed by probing immunoblots with antibodies for NMIIA, HER3 and α-tubulin.
(TIF)

**S2 Fig. Reduced total HER3 levels as a result of pharmacological inhibition of NMIIA in breast cancer cell lines. (A)** BT474 and MDA-MB-453 cells were treated with DMSO, neratinib, blebbistatin, and combination therapy of neratinib and blebbistatin for 24 hours. RNA was isolated and MYH9 and HER3 RNA levels determined using real time PCR. Graphs were blotted in Prism 7 (GraphPad) **(B)** BT474 and MDA-MB-453 cells were treated with DMSO, neratinib, blebbistatin, and combination therapy of neratinib and blebbistatin for 24 hours. Cells were lysed in RIPA buffer and the products were analyzed by 10% SDS-PAGE followed by probing immunoblots with antibodies against NMIIA, HER3, and α-tubulin.
(TIF)

**S3 Fig. Evaluation of colony formation and proliferation with doxycycline and neratinib treatment in HER2+ breast cancer cells expressing shRNA non-targeting control. (A)** BT474 and MDA-MB-453 cells transduced with non-targeting control shRNA were seeded on a basement membrane of matrigel at a density of $2 \times 10^4$ cells/well. Cells treated with vehicle (DMSO), doxycycline (500 nM)/48 h, neratinib (200 nM) once, and combination. Phase

contrast images of acini for cell lines were captured at 10× magnification and the average size of each cellular structure was quantified and expressed as mean of areas ± SEM, n = 5 random fields (**B**). (**C**) BT474 and MDA-MB-453 cells transduced with a doxycycline inducible shRNA targeting sh-control (shCTRL) were treated with doxycycline (500 nM) and neratinib (200 nM) for 72 hr. After 72 hr, the media containing the drug was replaced with 5 mg/mL MTT dissolved in cell line specific media and incubated for 4 hr. After 4 hr the media was aspirated, and crystals were dissolved with isopropanol. The bar graphs are represented as mean; Error bars: SEM, n = 3.
(TIF)

**S4 Fig. MYH9 knockdown inhibits spheroid growth of BT474 cells.** 1000 cells expressing shRNA targeting MYH9 or cell expressing non-targeting control (shCTRL) were plated in 96-well Ultra-Low Attachment plates. Cells were treated upon plating and every 48 hours with 500 nM doxycycline or vehicle. Images of spheroids were taken 10x magnification (left panel). PrestoBlue reagent was added on days 3, 5, and 7 to assess cell viability. Data shown is normalized to the values for vehicle treatment. The bar graphs are represented as mean; Error bars: SEM, n = 3. * p <0.05 for comparison to vehicle (right panel).
(TIF)

**S5 Fig. Evaluation of migration and invasion with doxycycline and neratinib treatment in HER2+ breast cancer cells expressing shRNA non-targeting control. (A)** BT474 and MDA-MB-453 ($5 \times 10^4$ cells/well) stable cells transduced with non-targeting control shRNA were seeded in six well plates and treated with vehicle (DMSO), doxycycline (500 nM), neratinib (200 nM) and indicated combination in serum free media for 24 h. Post 24 hr treatment, these treated cells ($2 \times 10^4$ cells/well) were added to the upper chamber of transwell plates with 10% FBS as chemoattractant in the lower chamber. After 24h, the migrated cells were stained and captured. **(B)** The intensities of the migrated cells were measured using ImageJ and expressed as mean of % control and represented as bar graph. Error bars: SEM, n = 4 **(C)** Invasion assays were performed where each chamber was coated with 20 <l matrigel mixed with 80 <l serum free media and incubated at 37˚ for 30 mins. Stable cells (BT474 and MDA-MB-453) expressing non-targeting control shRNA were seeded at a density of $5 \times 10^4$ cells/well and treated and incubated as above. After 24h, the invaded cells were stained and photographed **(D)** The intensities of the invaded cells were measured using ImageJ and expressed as mean % of control. Error bars: SEM, n = 3.
(TIF)

**S1 Raw images.**
(PDF)

## Acknowledgments

The protein identification by mass spectrometry was done in the UC Proteomics Laboratory under the direction of Ken Greis, PhD with subsidized recharge rates through support from the University of Cincinnati College of Medicine, Office of Research.

## Author Contributions

**Conceptualization:** Samar M. Alanazi, Sarah J. Storr, Joan T. Garrett.

**Data curation:** Samar M. Alanazi, Wasim Feroz, Rosalin Mishra, Hima Patel, Long Yuan, Sarah J. Storr.

**Formal analysis:** Wasim Feroz, Sarah J. Storr.

**Investigation:** Samar M. Alanazi, Sarah J. Storr.

**Methodology:** Samar M. Alanazi, Rosalin Mishra, Mary Kate Kilroy, Sarah J. Storr.

**Project administration:** Joan T. Garrett.

**Resources:** Joan T. Garrett.

**Validation:** Samar M. Alanazi.

**Writing – original draft:** Samar M. Alanazi, Sarah J. Storr.

**Writing – review & editing:** Samar M. Alanazi, Rosalin Mishra, Joan T. Garrett.

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
