## [Decision Letter · Decision Letter 0]

16 Aug 2022

PONE-D-22-17109HER2 Inhibition Increases Non-Muscle Myosin IIa to Promote Tumorigenesis in HER2+ Breast CancersPLOS ONE

Dear Dr. Joan Garrett

Thank you for submitting your manuscript to PLOS ONE. After careful consideration, we feel that it has merit but does not fully meet PLOS ONE’s publication criteria as it currently stands. Therefore, we invite you to submit a revised version of the manuscript that addresses the points raised during the review process.

We look forward to receiving your revised manuscript.

Kind regards,

Mohammed Soutto, Ph.D.

Academic Editor

PLOS ONE

Journal Requirements:

- https://aacrjournals.org/cancerres/article/82/4_Supplement/P5-10-05/681281/Abstract-P5-10-05-HER2-inhibition-increases-non

In your revision ensure you cite all your sources (including your own works), and quote or rephrase any duplicated text outside the methods section. Further consideration is dependent on these concerns being addressed.

Reviewers' comments:

Reviewer's Responses to Questions

**Comments to the Author**

1. Is the manuscript technically sound, and do the data support the conclusions?

Reviewer #1: Yes

Reviewer #2: Yes

2. Has the statistical analysis been performed appropriately and rigorously? 

Reviewer #1: Yes

Reviewer #2: I Don't Know

3. Have the authors made all data underlying the findings in their manuscript fully available?

Reviewer #1: Yes

Reviewer #2: Yes

4. Is the manuscript presented in an intelligible fashion and written in standard English?

Reviewer #1: Yes

Reviewer #2: Yes

5. Review Comments to the Author

Reviewer #1: In this manuscript entitled “HER2 Inhibition Increases Non-Muscle Myosin IIa to Promote Tumorigenesis in HER2+ Breast Cancers”, the authors showed that HER2 inhibition upregulates HER3 which binds to NMIIA and promotes breast cancer progression.

Overall, the manuscript is interesting and well written but additional edits and adjustments are required to strengthen the conclusions and improve the overall quality of the paper.

1) All figures are blurry and hard to read even the full-size image. Authors should provide higher resolution figures.

2) In line 75, the authors may have incorrectly cited reference 19. It should be NMIIA inhibition resulted in decreased cell migration. The way it is written shows opposite meaning.

3) In Figure 1A, did the authors try 200nM neratinib treatment? Do you also see an increase in NMIIA with this dose? The authors are advised to provide a better resolution of the figure.

4) In Figures 1B and C, a suggestion to the authors to use actin or GAPDH to normalize the input and can also use the heavy chain in IP samples to normalize loading. This will make it easier to interpret the results and to better judge if the treatment is enhancing biding or HER3 and NMIIA.

5) In Figure 3B, p-HER2 and HER2 should be added to the blots as a confirmation of inhibition with treatment.

6) In Figure 4B, HER2 inhibition is said to upregulate HER3 transcription and phosphorylation; however, there is no increase in p-HER3 with neratinib, only total is slightly increasing and mainly in BT474 cells; no increase in phosphorylated or total HER3 in MD-MB-453 cells with neratinib treatment. It seems BT474 are more responsive. Is this phosphorylated HER3 an activated form of HER3? This needs to be explained.

Another concern in the MDA-MB-453 blot, the last lane (shMYH9+Dox+Ner) doesn’t show knockdown of NMIIA yet there is an effect on downstream targets. This needs to be addressed.

7) In Figures 5 and 6, the quantification for sh-control and sh-MYH9 need to be combined in one graph and the comparison should be made between sh-control and sh-MYH9 with and without neratinib to see if the combination is really making a statistically significant difference. Please revise.

Please provide better resolution for the images, they are blurry and it is really hard to see the colonies or the invading/migrating cells and to read the labels.

8) Minor: in lines 222 and 229, “affect” should be replaced by “effect”

Reviewer #2: Title of study: HER2 Inhibition Increases Non-Muscle Myosin IIa to Promote Tumorigenesis in

HER2+ Breast Cancers

This study focuses on the combination therapy for HER3+ breast cancers. The authors are proposing that MYH9 knockdown along with HER2+ inhibition provides better treatment for HER2+ breast cancers. The authors show that this combination treatment involves HER3 which directly regulates the Non-muscle Myosin II-A (NMIIA) levels. First, they show that inhibiting HER2+ with Neratinib, a well-studied and reported tyrosine kinase inhibitor of HER2+, results in increased protein and mRNA levels of NMIIA and HER3 in two breast cancer cell lines. Next, they determine through Immunoprecipitation that HER3 and NMIIA bind to each other after treatment with Neratinib. Also, they stably knockdown NNMIIA in these breast cancer cell lines by lentivirus targeting the gene MYH9 which codes for NMIIA. Consequently, they show that treatment of shMYH9 cancer cells with Neratinib results in more significant decreased phosphorylation of HER3 and the downstream targets AKT and ERK1/2 when compared to Neratinib alone. Further, they perform colony formation and MTT assays and show that MYH9 knockdown along with Neratinib treatment inhibit colony formation and proliferation more significantly. These results are further confirmed through Migration/Invasion assays which demonstrate that migration and invasion of cells is greatly reduced after the Neratinib treatment in shMYH9 breast cancer cells compared to Neratinib alone.

All these results support the aim of the study but there are numerous shortcomings as well that need to be addressed.

Minor comments:

Although there are not many typos in this manuscript but still needs some overview to correct some, like in Section 3.4 ‘affect’ can be replaced by ‘effect’.

‘Figure 2’ in Results section must be in bold format.

In Figure 2, make the legend names the same color as in the graph to make it easy to read.

Major comments:

The figures are not in a readable format at all because of the extreme blurred focus. The resolution needs to be fixed to make it visible and clear to read and interpret the results.

Results have been repeated in Discussion section. Discussion should be re-written, and the content should be different from the results section.

The statistical analysis seems to be not correct based on the error bars especially in Figure 3, could you please provide a table or the raw data shown in this figure and also in the rest of the figures that have bars graph and the errors are not clear because of the poor resolution of the figures.

Figure 1:

In figure 1 panel 1(c), the pulldown of HER3 also shows band in DMSO Control when compared to the Input. Can authors comment on that?

Figure 3:

In figure 3(A), Control DMSO treatment is missing the error bars and please mention how many times was the experiments repeated and reproduced.

Figure 4:

In figure 4(A) the control sample shows high protein levels of NMIIA but the same cells in Figure 3 show no or very less protein levels of NMIIA. Why were these cells used to knockdown NMIIA (MYH9 gene) when they don’t express NMIIA endogenously based on results from figure 3?

Figure 5 and 6:

These figures are very small and hardly visible, please fix the resolution.

The authors need to show quantification of MTT assay and Migration/Invasion results all in one graph to compare results between shRNA Ctrl and shRNA MYH9 results.

6. PLOS authors have the option to publish the peer review history of their article (what does this mean?). If published, this will include your full peer review and any attached files.

Reviewer #1: **Yes: **Farah Ballout

Reviewer #2: No

---

## [Author Response · Author response to Decision Letter 0]

15 Jan 2023

Re: PONE-D-22-17109

Title: HER2 Inhibition Increases Non-Muscle Myosin IIa to Promote Tumorigenesis in HER2+ Breast Cancers

Dear PLOS One Editorial Board,

Please find enclosed the revised manuscript. We sincerely appreciate the critique and input from reviewers as well as the opportunity you give us to address concerns and resubmit our manuscript. We will address below issues/concerns raised in the Critique. A marked-up copy of the changes with highlighting in yellow made from the previous article can be found under 'Revised Manuscript with Track Changes' file. The text highlighted yellow in this letter are the sentences from the manuscript that have been modified to reflect the changes as per reviewer suggestions. 

Journal Requirements:

We thank the journal for pointing this out. We have changed all major sections to level 1 heading, bold type, 18pt font. We have added the figure captions to the body of the text, directly after the paragraph in which they are cited. We have added the supporting information captions at the end of the manuscript in a section titled “Supporting Information”.

- https://aacrjournals.org/cancerres/article/82/4_Supplement/P5-10-05/681281/Abstract-P5-10-05-HER2-inhibition-increases-non

In your revision ensure you cite all your sources (including your own works), and quote or rephrase any duplicated text outside the methods section. Further consideration is dependent on these concerns being addressed.

We thank the journal for this feedback. We have significantly altered the abstract to match with the inclusion of new data and rephrased duplicated text from the abstract of the 2021 SABCS Annual Meeting so that there should not be overlapping text.

We have included original uncropped and unadjusted images underlying all blot results in a file titled S1_raw_images. This file is uploaded as a Supporting Information file.

We thank the journal for this feedback and have included captions for Supporting Information files at the end of the manuscript.

Reviewer comments:

Reviewer #1: In this manuscript entitled “HER2 Inhibition Increases Non-Muscle Myosin IIa to Promote Tumorigenesis in HER2+ Breast Cancers”, the authors showed that HER2 inhibition upregulates HER3 which binds to NMIIA and promotes breast cancer progression.

Overall, the manuscript is interesting and well written but additional edits and adjustments are required to strengthen the conclusions and improve the overall quality of the paper.

1) All figures are blurry and hard to read even the full-size image. Authors should provide higher resolution figures.

We sincerely apologize for the low-resolution images and are thankful to the reviewer for this comment. Accordingly, we have uploaded high resolution images for figures.

2) In line 75, the authors may have incorrectly cited reference 19. It should be NMIIA inhibition resulted in decreased cell migration. The way it is written shows opposite meaning.

We thank the reviewer for noting this error. We have closely examined reference 19 and edited the text to the following:

A study in MDA-MB-231 breast cancer cells showed that depletion of NMIIA with small-interfering RNA resulted in decreased MDA-MB-231 cell migration but enhanced lamellar spreading [1].

3) In Figure 1A, did the authors try 200nM neratinib treatment? Do you also see an increase in NMIIA with this dose? The authors are advised to provide a better resolution of the figure.

To initially identify HER3 binding partners that may be differentially expressed upon HER2 inhibition, we used 500 nM neratinib. Further in our studies through personal communications with Puma Biotechnology who developed neratinib, we focused on 200 nM neratinib which is more clinically achievable in patients. Although we did not repeat the original experiment of Figure 1A with 200 nM neratinib, subsequent experiments with 200 nM neratinib resulted in increased NMIIA relative to vehicle control (Figure 1B, C, Figure 3).

4) In Figures 1B and C, a suggestion to the authors to use actin or GAPDH to normalize the input and can also use the heavy chain in IP samples to normalize loading. This will make it easier to interpret the results and to better judge if the treatment is enhancing biding or HER3 and NMIIA.

We thank the reviewer for this comment and have now included a loading control of α-tubulin for the input samples in Figures 1B and 1C.

5) In Figure 3B, p-HER2 and HER2 should be added to the blots as a confirmation of inhibition with treatment.

We are grateful to the reviewer for this comment. Accordingly, we have added p-HER2 and HER2 blots in Figure 3B as confirmation of inhibition HER2 kinase activity. 

6) In Figure 4B, HER2 inhibition is said to upregulate HER3 transcription and phosphorylation; however, there is no increase in p-HER3 with neratinib, only total is slightly increasing and mainly in BT474 cells; no increase in phosphorylated or total HER3 in MD-MB-453 cells with neratinib treatment. It seems BT474 are more responsive. Is this phosphorylated HER3 an activated form of HER3? This needs to be explained.

Another concern in the MDA-MB-453 blot, the last lane (shMYH9+Dox+Ner) doesn’t show knockdown of NMIIA yet there is an effect on downstream targets. This needs to be addressed.

We thank the reviewer and apologize for the poor wording in the previous version of our manuscript. HER2 inhibition can indeed upregulate HER3 transcription and protein levels while inhibiting HER2 kinase. Inhibiting HER2 kinase results in inhibition of HER3 phosphorylation. At later time points there can be a recovery of phosphorylated HER3 over time using a HER2 inhibitor [2]. For clarity we have edited the abstract to state: 

Increased HER3 transcription and protein levels occur upon inhibition of HER2. 

Phosphorylated Y1289 of HER3 is an activated form of HER3. For clarity we have added the following text to the manuscript:

Y1289 of HER3 resides within a YXXM motif and participates in signaling to PI3K upon phosphorylation, activating downstream signaling [3].

For the MDA-MB-453 blot examining NMIIA protein level, we quantified band intensities using ImageJ software. Normalizing intensities to loading control α-tubulin, the relative band intensity of shCTRL (lane 1) is 1.0, shMYH9 + Dox (lane 3) is 0.15 and shMYH9 + Dox + Ner (lane 6) 0.69. We agree that the shRNA targeting NMIIA is less effective with neratinib treatment, however this is still a ~30% reduction in NMIIA relative to lane 1.

7) In Figures 5 and 6, the quantification for sh-control and sh-MYH9 need to be combined in one graph and the comparison should be made between sh-control and sh-MYH9 with and without neratinib to see if the combination is really making a statistically significant difference. Please revise.

Please provide better resolution for the images, they are blurry and it is really hard to see the colonies or the invading/migrating cells and to read the labels.

We thank the reviewer for this feedback. Upon further consideration we have generated distinct cells lines expressing either shCTRL or shMYH9. These cell lines were treated with puromycin for different time points as the lentiviral integration and incorporation of puromycin resistance gene varied between the lentiviral constructs. As such each of these distinct cell lines has growth properties that vary between shCTRL and shMYH9 in the absence of doxycycline. It would be inappropriate to combine distinct cell lines into one graph. For clarity we have moved shCTRL cell line data into supplemental files.

8) Minor: in lines 222 and 229, “affect” should be replaced by “effect”

We are grateful to the reviewer for finding this error. We have corrected the manuscript.

Reviewer #2: Title of study: HER2 Inhibition Increases Non-Muscle Myosin IIa to Promote Tumorigenesis in

HER2+ Breast Cancers. This study focuses on the combination therapy for HER3+ breast cancers. The authors are proposing that MYH9 knockdown along with HER2+ inhibition provides better treatment for HER2+ breast cancers. The authors show that this combination treatment involves HER3 which directly regulates the Non-muscle Myosin II-A (NMIIA) levels. First, they show that inhibiting HER2+ with Neratinib, a well-studied and reported tyrosine kinase inhibitor of HER2+, results in increased protein and mRNA levels of NMIIA and HER3 in two breast cancer cell lines. Next, they determine through Immunoprecipitation that HER3 and NMIIA bind to each other after treatment with Neratinib. Also, they stably knockdown NNMIIA in these breast cancer cell lines by lentivirus targeting the gene MYH9 which codes for NMIIA. Consequently, they show that treatment of shMYH9 cancer cells with Neratinib results in more significant decreased phosphorylation of HER3 and the downstream targets AKT and ERK1/2 when compared to Neratinib alone. Further, they perform colony formation and MTT assays and show that MYH9 knockdown along with Neratinib treatment inhibit colony formation and proliferation more significantly. These results are further confirmed through Migration/Invasion assays which demonstrate that migration and invasion of cells is greatly reduced after the Neratinib treatment in shMYH9 breast cancer cells compared to Neratinib alone.

All these results support the aim of the study but there are numerous shortcomings as well that need to be addressed.

Minor comments: Although there are not many typos in this manuscript but still needs some overview to correct some, like in Section 3.4 ‘affect’ can be replaced by ‘effect’.

We are grateful to the reviewer for pointing out these errors. We have corrected the manuscript.

‘Figure 2’ in Results section must be in bold format.

We apologize for the lack of consistency in formatting. We have updated the manuscript with consistent formatting for descriptions of figures.

In Figure 2, make the legend names the same color as in the graph to make it easy to read.

This is an excellent suggestion. We have completely revised Figure 2 to make a stronger justification for studying NMIIA in breast cancer utilizing breast cancer clinical data, particularly in HER2+ breast cancer. In the revised figure, we include information in grayscale.

Major comments: The figures are not in a readable format at all because of the extreme blurred focus. The resolution needs to be fixed to make it visible and clear to read and interpret the results.

We sincerely apologize for this oversight and have included high resolution figures in the revision.

Results have been repeated in Discussion section. Discussion should be re-written, and the content should be different from the results section.

We thank the reviewer for this feedback and have closely revised the discussion section. We have copied below edits made which are highlighted in yellow. 

Recent work to delineate how NMII isoforms function upon a stiffening extracellular matrix in cancer cells found that NMIIA aides the establishment of cell polarity through actin polarization at the cell leading edge, lamellipodia formation and focal adhesion turnover [4].

Conversely, other studies have shown that NMIIA can act as a tumor suppressor. Repression of MYH9 predisposed mice to squamous cell carcinoma (SCC), as MYH9 was identified from an RNA interference screen [5]. The mechanism by which NMIIA acts as a tumor suppressor was via a noncanonical role in nuclear retention of activated p53. When NMIIA is defective, even though the p53 pathway can be induced in response to DNA damage, it did not occur because of an inability of p53 to remain and/or accumulate in the nucleus. Additional work found that low MYH9 expression is associated with decreased survival in patients with head and neck cancer harboring stratified low-risk mutant p53 but not high-risk mutant p53 [6]. More than half of all human cancers contain a TP53 mutation. Notably BT474 cells have been reported to have a p53 mutation and MDA-MB-453 cells have a deletion in p53 [7]. As the breast cancer models in our study may not depend on functional p53, the role of NMIIA in nuclear retention of activated p53 may be irrelevant for the breast cancer models in our study. Mice with homozygous deletion of NMIIA consistently developed SCC of the tongue [8]. Interestingly genetic deletion of NMIIA did not affect the stability of p53 but was ascribed to NMIIA maintaining mitotic stability during cell division. An additional study found that genetically knocking down MYH9 in mouse melanoma cells increased growth and metastasis in vivo in mouse models which was attributed to increased epithelial to mesenchymal (EMT) markers and potentially increased angiogenesis and inflammatory cell recruitment [9]. These studies point to the context in which NMIIA may be needed throughout development and the interplay with the tumor microenvironment in determining if NMIIA permits or hampers tumor growth. 

Blebbistatin reversibly inhibits NMII by blocking its ATPase activity [10-13]. Due to blebbistatin’s non-specificity and cytotoxicity [14], it may not be a good candidate to target NMIIA in HER2+ breast cancer. Recently however blebbistatin has been used in vivo in tumorgenicity studies demonstrating reduced tumor volume in a dose-dependent manner in a NSCLC model [15]. Toxicity or weight loss with blebbistatin in mice was not reported although the lack of specificity for blebbistatin is a major consideration. Blebbistatin has been reported to inhibit NMIIA and NMIIB along with several striated muscle myosins and data indicate that blebbistatin inhibits arterial smooth muscle tonic contractions [16, 17]. There is a strong rationale for the development of more specific inhibitors of NMIIA that could be used in cancer treatment, potentially in HER2+ breast cancers from data presented in this study.

Recently, Orgaz et al. demonstrated that myosin II activity increases during melanoma adaptation to MAPK pathway inhibition. High myosin II activity correlated with targeted and immunotherapy resistant melanomas [18]. Rho-kinase (ROCK) is upstream of NMIIA. ROCK inactivates the myosin light chain 2 (MLC2) phosphatase, which leads to increased phosphorylation of MLC2 (p-MLC2) and myosin II activity. ROCK inhibitors improve the efficacy of MAPK inhibitors and immunotherapies in melanoma models [18]. The utility of ROCK inhibitors should be considered in combination with HER2 inhibition in HER2+ breast cancers given the paucity of NMIIA specific inhibitors.

We observed a bidirectional relationship between NMIIA and the HER family. Fig 3 demonstrates that inhibition of HER signaling with neratinib treatment increases MYH9 mRNA levels and the protein NMIIA. Future studies will define the mechanistic details of neratinib induced MYH9 upregulation. There are limitations to our work. This study does not determine mechanistically how HER3 and NMIIA interact nor the mechanism(s) by which loss of NMIIA reduces HER3 protein levels or how the loss of HER3 reduces NMIIA protein levels. The immunoprecipitation experiments shown in Fig 1B, C are challenging to interpret. The immunoprecipitation of HER3 followed by immunoblotting of HER3 reveals unequal amounts of HER3 between lanes, similarly for immunoprecipitation of NMIIA followed by immunoblotting of NMIIA. As shown in Fig 3, 24 hours of neratinib treatment results in increased NMIIA and HER3, correlating with the increased amount of HER3 and NMIIA detected from the immunoprecipitation and immunoblotting experiments shown in Fig 1B, C. Future studies will mechanistically decipher which compartment(s) of the cell NMIIA and HER3 interact and if blebbistatin can inhibit the interaction. Future studies will delineate if loss of MYH9 affects the half-life of HER3 protein and examine the effects of overexpression of NMIIA.

The statistical analysis seems to be not correct based on the error bars especially in Figure 3, could you please provide a table or the raw data shown in this figure and also in the rest of the figures that have bars graph and the errors are not clear because of the poor resolution of the figures.

Figure 1: In figure 1 panel 1(c), the pulldown of HER3 also shows band in DMSO Control when compared to the Input. Can authors comment on that?

We thank the reviewer for the comment. We performed many iterations to optimize immunoprecipitation experiments which are challenging. To get a signal for immunoprecipitation experiments, we used 1400 µg of protein lysate to immunoprecipitate HER3. For the input samples, approximately 15 µg of protein lysate was used, nearly 1000-fold less protein from immunoprecipitation experiments, making detection of the input lanes less sensitive.

Figure3: In figure 3(A), Control DMSO treatment is missing the error bars and please mention how many times was the experiments repeated and reproduced.

We thank the reviewer for this comment. All treatment groups including control DMSO were performed in triplicate and repeated for a total of three independent experiments unless indicated otherwise. Candidly, we have asked the first author who performed experiments to provide the source files. In September, she emailed nearly all the files but not the qPCR files. Setting vehicle DMSO values to 1, error bars were inadvertently excluded. The first author of the manuscript has metastatic cancer and has frequently been hospitalized since September with blood stream infections, COVID and procedures related to shrinking lesions in her lungs. Due to this we are unable to update Figure 3A to include error bars for vehicle DMSO. However, the statistical analysis is still valid comparing triplicate samples using a two-sample paired ‘t’-test.

Figure4: In figure 4(A) the control sample shows high protein levels of NMIIA but the same cells in Figure 3 show no or very less protein levels of NMIIA. Why were these cells used to knockdown NMIIA (MYH9 gene) when they don’t express NMIIA endogenously based on results from figure 3?

We thank the reviewer for this keen observation. Figure 3 shows parental BT474 and MDA-MB-453 cells. Figure 4 shows cell lines that have been transduced with lentivirus containing shRNA. These cell lines were then treated with puromycin to select for cells that have incorporation of the puromycin resistance gene. As these are separate cell lines between Figure 3 and Figure 4A, direct comparisons cannot be made. Our studies focused on HER2+ breast cancers based on evidence that NMIIA is clinically relevant in HER2+ breast cancer (Table 1). We acknowledge that NMIIA is expressed at relative lower levels in MDA-MB-453 cells based on immunoblotting analysis. We examined a panel of HER2+ breast cancer cell lines for (1) the ability for neratinib to increase NMIIA expression and (2) for NMIIA to bind HER3 upon neratinib treatment based on immunoprecipitation experiments as shown in Figure 1A. MDA-MB-453 cell lines showed these properties as shown in Figure 1C, Figure 3 and therefore we focused on this cell line in the manuscript.

Figure 5 and 6: These figures are very small and hardly visible, please fix the resolution.

We sincerely apologize for the low-resolution of these images and have included high resolution images in the revised submission.

The authors need to show quantification of MTT assay and Migration/Invasion results all in one graph to compare results between shRNA Ctrl and shRNA MYH9 results.

We thank the reviewer for this feedback. Upon further consideration we have generated distinct cells lines expressing either shCTRL or shMYH9. These cell lines were treated with puromycin for different time points as the lentiviral integration and incorporation of puromycin resistance gene varied between the lentiviral constructs. As such each of these distinct cell lines has growth properties that vary between shCTRL and shMYH9 in the absence of doxycycline. It would be inappropriate to combine distinct cell lines into one graph. For clarity we have moved shCTRL cell line data into supplemental files.

We hope that this rebuttal, changes, and additions have improved our paper to a point in which it is acceptable for reevaluation by the Editorial Board of PLOS One. Thank you very much for your time and consideration.

Sincerely,

Joan T. Garrett

1. Betapudi, V., L.S. Licate, and T.T. Egelhoff, Distinct roles of nonmuscle myosin II isoforms in the regulation of MDA-MB-231 breast cancer cell spreading and migration. Cancer research, 2006. 66(9): p. 4725-4733.

2. Garrett, J.T., et al., Transcriptional and posttranslational up-regulation of HER3 (ErbB3) compensates for inhibition of the HER2 tyrosine kinase. Proc Natl Acad Sci U S A, 2011. 108(12): p. 5021-6.

3. Kim, H.H., S.L. Sierke, and J.G. Koland, Epidermal growth factor-dependent association of phosphatidylinositol 3-kinase with the erbB3 gene product. Journal of Biological Chemistry, 1994. 269(40): p. 24747-24755.

4. Peng, Y., et al., Non-muscle myosin II isoforms orchestrate substrate stiffness sensing to promote cancer cell contractility and migration. Cancer Lett, 2022. 524: p. 245-258.

5. Schramek, D., et al., Direct in vivo RNAi screen unveils myosin IIa as a tumor suppressor of squamous cell carcinomas. Science, 2014. 343(6168): p. 309-13.

6. Coaxum, S.D., et al., The tumor suppressor capability of p53 is dependent on non-muscle myosin IIA function in head and neck cancer. Oncotarget, 2017. 8(14): p. 22991-23007.

7. Runnebaum, I.B., et al., Mutations in p53 as potential molecular markers for human breast cancer. Proc Natl Acad Sci U S A, 1991. 88(23): p. 10657-61.

8. Conti, M.A., et al., Conditional deletion of nonmuscle myosin II-A in mouse tongue epithelium results in squamous cell carcinoma. Sci Rep, 2015. 5: p. 14068.

9. Singh, S.K., et al., MYH9 suppresses melanoma tumorigenesis, metastasis and regulates tumor microenvironment. Med Oncol, 2020. 37(10): p. 88.

10. Roman, B.I., S. Verhasselt, and C.V. Stevens, Medicinal chemistry and use of myosin II inhibitor (S)-blebbistatin and its derivatives. Journal of medicinal chemistry, 2018. 61(21): p. 9410-9428.

11. Straight, A.F., et al., Dissecting temporal and spatial control of cytokinesis with a myosin II Inhibitor. Science, 2003. 299(5613): p. 1743-7.

12. Kovacs, M., et al., Mechanism of blebbistatin inhibition of myosin II. J Biol Chem, 2004. 279(34): p. 35557-63.

13. Allingham, J.S., R. Smith, and I. Rayment, The structural basis of blebbistatin inhibition and specificity for myosin II. Nat Struct Mol Biol, 2005. 12(4): p. 378-9.

14. Rauscher, A.Á., et al., Targeting myosin by blebbistatin derivatives: optimization and pharmacological potential. Trends in biochemical sciences, 2018. 43(9): p. 700-713.

15. Chen, M., et al., MYH9 is crucial for stem cell-like properties in non-small cell lung cancer by activating mTOR signaling. Cell Death Discov, 2021. 7(1): p. 282.

16. Eddinger, T.J., et al., Potent inhibition of arterial smooth muscle tonic contractions by the selective myosin II inhibitor, blebbistatin. J Pharmacol Exp Ther, 2007. 320(2): p. 865-70.

17. Limouze, J., et al., Specificity of blebbistatin, an inhibitor of myosin II. J Muscle Res Cell Motil, 2004. 25(4-5): p. 337-41.

18. Orgaz, J.L., et al., Myosin II Reactivation and Cytoskeletal Remodeling as a Hallmark and a Vulnerability in Melanoma Therapy Resistance. Cancer Cell, 2020. 37(1): p. 85-103 e9.

---

## [Decision Letter · Decision Letter 1]

19 Jan 2023

PONE-D-22-17109R1

HER2 Inhibition Increases Non-Muscle Myosin IIa to Promote Tumorigenesis in HER2+ Breast Cancers

PLOS ONE

Dear Dr. Garrett,

Thank you for submitting your manuscript to PLOS ONE. After careful consideration, we feel that it has merit but does not fully meet PLOS ONE’s publication criteria as it currently stands. Therefore, we invite you to submit a revised version of the manuscript that addresses the points raised during the review process.

We look forward to receiving your revised manuscript.

Kind regards,

Mohammed Soutto, Ph.D.

Academic Editor

PLOS ONE

Journal Requirements:

Reviewers' comments:

Reviewer's Responses to Questions

**Comments to the Author**

1. If the authors have adequately addressed your comments raised in a previous round of review and you feel that this manuscript is now acceptable for publication, you may indicate that here to bypass the “Comments to the Author” section, enter your conflict of interest statement in the “Confidential to Editor” section, and submit your "Accept" recommendation.

Reviewer #1: All comments have been addressed

Reviewer #2: All comments have been addressed

2. Is the manuscript technically sound, and do the data support the conclusions?

Reviewer #1: Yes

Reviewer #2: Yes

3. Has the statistical analysis been performed appropriately and rigorously? 

Reviewer #1: Yes

Reviewer #2: needs to be clarified

4. Have the authors made all data underlying the findings in their manuscript fully available?

Reviewer #1: Yes

Reviewer #2: Yes

5. Is the manuscript presented in an intelligible fashion and written in standard English?

Reviewer #1: Yes

Reviewer #2: Yes

6. Review Comments to the Author

Reviewer #1: (No Response)

Reviewer #2: The questions have fixed the concerns raised in their manuscript. The Discussion part has been re-written and updated with new findings. All the questions have been addressed except for the statistical analysis in Fig 3. The authors need to mention which stats have been used to calculate the significance.

As previously mentioned in the first review, there was no error bar on DMSO controls in Fig 3, the question has not been addressed fully. As mentioned in the response that the first author is critically sick and undergoing treatment, I fully sympathize with the author's condition and hope for her complete recovery. However, it would be solid if this figure 3 was repeated to ensure three individual experiments were done so that there remain no concerns about the results in this figure.

7. PLOS authors have the option to publish the peer review history of their article (what does this mean?). If published, this will include your full peer review and any attached files.

Reviewer #1: No

Reviewer #2: **Yes: **Nadeem Bhat

---

## [Author Response · Author response to Decision Letter 1]

13 Apr 2023

April 13, 2023

Re: PONE-D-22-17109

Title: HER2 Inhibition Increases Non-Muscle Myosin IIa to Promote Tumorigenesis in HER2+ Breast Cancers

Dear PLOS One Editorial Board,

Please find enclosed the revised manuscript. We sincerely appreciate the critique and input from reviewers as well as the opportunity you give us to address concerns and resubmit our manuscript. We will address below issues/concerns raised in the Critique. A marked-up copy of the changes with indicated under track changes from the previous article can be found under 'Revised Manuscript with Track Changes' file. The text highlighted yellow in this letter are the sentences from the manuscript that have been modified to reflect the changes as per reviewer suggestions. 

Reviewer comments:

Reviewer #2: The questions have fixed the concerns raised in their manuscript. The Discussion part has been re-written and updated with new findings. All the questions have been addressed except for the statistical analysis in Fig 3. The authors need to mention which stats have been used to calculate the significance.

As previously mentioned in the first review, there was no error bar on DMSO controls in Fig 3, the question has not been addressed fully. As mentioned in the response that the first author is critically sick and undergoing treatment, I fully sympathize with the author's condition and hope for her complete recovery. However, it would be solid if this figure 3 was repeated to ensure three individual experiments were done so that there remain no concerns about the results in this figure.

We thank the reviewer for the comments. We agree that Figure 3A of the previous version of the manuscript was lacking in scientific rigor. As such, we have worked to independently repeat qRT-PCR results. We apologize for the delay in response, as we worked to optimize use with a new qPCR machine and subsequently perform a minimum of three biological replicates. In performing these assays, we did not find that neratinib treatment significantly modulated MYH9 mRNA levels. As such, we have removed Figure 3A of the previous manuscript. These findings indicate that HER2 inhibition increases NMIIA levels at the post-translational level and not at the transcriptional level. We have edited the text to the following:

HER3 and NMIIA protein levels are increased upon inhibition of HER2 in HER2+ breast cancer

We next sought to examine overall levels of HER3 and MYH9 mRNA and protein upon treatment with neratinib in HER2+ breast cancer cell lines. Levels of HER3 and MYH9 mRNA in BT474 and MDA-MB-453 were assessed using qRT-PCR. We did not observe that mRNA levels of HER3 or MYH9 consistently changed upon HER2 inhibition with neratinib (data not shown). We found that 24 hours of neratinib treatment reduced P-HER2 and concomitant reduction in total HER2 protein as has been reported [1, 2]. We found that neratinib treatment increased HER3 and NMIIA protein levels (Fig 3).

We hope that this rebuttal, changes, and additions have improved our paper to a point in which it is acceptable for reevaluation by the Editorial Board of PLOS One. Thank you very much for your time and consideration.

Sincerely,

Joan T. Garrett

1. Collins, D.M., et al., Tyrosine kinase inhibitors as modulators of trastuzumab-mediated antibody-dependent cell-mediated cytotoxicity in breast cancer cell lines. Cell Immunol, 2017. 319: p. 35-42.

2. Collins, D.M., et al., Preclinical Characteristics of the Irreversible Pan-HER Kinase Inhibitor Neratinib Compared with Lapatinib: Implications for the Treatment of HER2-Positive and HER2-Mutated Breast Cancer. Cancers (Basel), 2019. 11(6).

---

## [Editor Report · Decision Letter 2]

19 Apr 2023

HER2 Inhibition Increases Non-Muscle Myosin IIa to Promote Tumorigenesis in HER2+ Breast Cancers

PONE-D-22-17109R2

Dear Dr. Garrett

We’re pleased to inform you that your manuscript has been judged scientifically suitable for publication and will be formally accepted for publication once it meets all outstanding technical requirements.

Kind regards,

Mohammed Soutto, Ph.D.

Academic Editor

PLOS ONE
---

## [Editor Report · Acceptance letter]

10 May 2023

PONE-D-22-17109R2 

HER2 Inhibition Increases Non-Muscle Myosin IIa to Promote Tumorigenesis in HER2+ Breast Cancers 

Dear Dr. Garrett:

I'm pleased to inform you that your manuscript has been deemed suitable for publication in PLOS ONE. Congratulations! Your manuscript is now with our production department. 

Kind regards, 

on behalf of

Dr. Mohammed Soutto 

Academic Editor

PLOS ONE